# Preliminary Results of Subtalar Arthroereisis with Vulpius Procedure for Symptomatic Flatfoot in Patients with Type I Osteogenesis Imperfecta

**DOI:** 10.3390/ijerph18010067

**Published:** 2020-12-24

**Authors:** Cheng-Min Hsu, Sheng-Chieh Lin, Kuan-Wen Wu, Ting-Ming Wang, Jia-Feng Chang, Chia-Che Lee

**Affiliations:** 1Department of Orthopedic Surgery, Chang Gung Memorial Hospital Linkou Main Branch, Taoyuan 333, Taiwan; potter2369@gmail.com; 2Department of Orthopedic Surgery, Chung Shan Medical University Hospital, Taichung 402, Taiwan; Phoenix33343@gmail.com; 3Department of Orthopedic Surgery, National Taiwan University Hospital, Taipei 100, Taiwan; wukuanwen@gmail.com (K.-W.W.); dtorth76@yahoo.com.tw (T.-M.W.); 4Department of Internal Medicine, Shuang Ho Hospital, New Taipei City 235, Taiwan; 01508@km.eck.org.tw; 5Department of Internal Medicine, En Chu Kong Hospital, New Taipei City 237, Taiwan

**Keywords:** osteogenesis imperfecta, subtalar arthroereisis, symptomatic flatfoot, pes planovalgus, gastrocnemius recession

## Abstract

In this retrospective study, we aim to assess the safety and feasibility of adapting subtalar arthroereisis (SA) for type I osteogenesis imperfecta (OI) patients with symptomatic flatfoot. From December 2013 to January 2018, six type I OI patients (five girls and one boy, 12 feet) with symptomatic flexible flatfoot were treated with SA and the Vulpius procedure. All the patients were ambulatory and skeletally immature with failed conservative treatment and unsatisfactory life quality. The median age at the time of surgery was 10 years (range 5–11), and the median follow-up period was 55 months (range 33–83). All functional and radiographic parameters improved (*p* < 0.05) after the procedure at the latest follow-up. The median American Orthopaedic Foot and Ankle Society ankle-hindfoot scale improved from 68 (range 38–80) to 95 (range 71–97). All of the patients ambulated well without significant complications. The weight-bearing radiographs showed maintained correction of the tarsal bone alignment with intact bony surfaces adjacent to implants during the post-operative follow-up period. This is the very first study on symptomatic flatfoot in pediatric patients with type I OI. Our data suggest that SA is a potentially viable approach, as functional improvements and maintained radiographic correction without significant complication were observed.

## 1. Introduction

Osteogenesis imperfecta (OI) is a rare, hereditary, heterogeneous connective tissue disorder characterized by extreme bone fragility and soft tissue defect, with a prevalence of 3–7 per 100,000 births [1]. Secondary to ligamentous laxity and joint hypermobility, the nature of the disease makes flatfoot an common deformity, reported in 24–75% of OI patients [2]. By contrast, the prevalence of flatfeet is reported to be around 20–24% in the general population [3]. Recently, numerous mutations in collagen-related genes have been discovered to cause OI. However, limited study of the epidemiology of flatfoot deformity in different genome types has been reported. Although the natural history of flatfoot in Sillence type I patients, the milder form of OI, has not been well documented in the literature, severe cases were noted in our institute (Figure 1). For example, a 52-year-old woman with Sillence type I OI complained of pain and deformity of her bilateral hindfeet. She had suffered from pain and flatfoot deformity for approximately 20 years prior to her first visit. Gradually, the pain intensity increased despite persistent non-operative management. She developed difficulty in ambulation with the need for a wheelchair after reaching her fifties. Rocker bottom and hind foot valgus deformity developed.

Symptomatic flatfoot deformity in Sillence type I OI patients is often overlooked because of the relatively high functional abilities and normal appearance compared to the other severe types of OI [4]. Bratberg et al. [5] reported that one out of two OI cases remained painful with unsatisfactory results after subtalar extra-articular arthrodesis because of the poor architectural quality of the bone. Agarwal et al. [6] suggested that about one-fifth of patients with OI have the risk of developing a non-union at the sites of osteotomies. Considering limited experience and possible poor recovery, the operative indications for symptomatic flatfoot in OI patient remain to be clarified.

Subtalar arthroereisis (SA) is a non-fusion type procedure in which an implant is inserted into the sinus tarsi between the posterior and anterior subtalar facet joints. The implant expands the subtalar joint vertically, elevates the head of the talus, prevents the excessive pronation of the hindfoot, realigns the longitudinal arch of the foot and subsequently reduces the flatfoot deformity. The procedure corrects the alignment of flatfoot, especially on lateral tarso-metatarsal alignment and talo-navicular congruency, and maintains this correction in the short to the mid-term until skeletal maturity [7,8]. Previous systematic reviews [9,10,11] of SA showed less tissue trauma and faster recovery with comparable results in correcting flatfeet to other managements in general populations. Characterized by a non-fusion nature and less tissue injury, SA may benefit OI patients in dealing with flatfoot. The purpose of this study is to conduct a retrospective study for symptomatic flatfoot in Sillence Type I OI patients to assess the safety, feasibility and the clinical results of SA for this challenging problem.

## 2. Materials and Methods

### 2.1. Study Design

With the approval of the Research Ethics Committee from our institute (201909005RINB), we conducted a retrospective study to review the 35 OI patients with symptomatic flatfoot deformity in our institute. Skeletally matured patients, non-ambulators or non-Sillence-type-I patients were excluded. Six patients (five girls and one boy, 12 feet) with symptomatic flexible flatfoot treated with bilateral SA and Vulpius procedure between December 2013 and January 2018 in our institute were identified. All of them had failed conservative managements including orthotic insoles, physical therapy and activity modification for at least 6 months before the consideration for surgical treatment.

Owing to the relatively normal functional status, only two of the included six patients underwent genetic study. Both of them have mutations on COL1A1 gene and received regular bisphosphonate therapy. The others were diagnosed by pediatric geneticists according to a history of multiple fracture episodes, clinical presentations including blue sclerae, near-normal stature, lack of apparent dentinogenesis imperfecta and positive family history. The median age at the time of surgery was 10 years old (range 5–11). The median postoperative follow-up was 55 months (range 33–83). All patients who underwent surgery showed a positive toe raising test and Silfverskiöld test on bilateral feet, indicating flexible flatfoot and gastrocnemius tightness with little or no soleus involvement.

### 2.2. Operative Technique

The operations were done under general anesthesia with the use of a thigh tourniquet. The patient was placed in a supine position. The muscle–tendon junction around the middle and distal third of the calf was identified by palpation. A 1.5 cm longitudinal incision was made for the Vulpius procedure with an inverted “V” resection made through the aponeuroses of the gastrocnemius. Another small incision was made on the lateral aspect of the foot over the sinus tarsi area. It is important to avoid the intermediate dorsal cutaneous nerves and the sural nerve, which course superior and inferior to the incision, respectively, by making the incision parallel to the nerves. The cannulated probe was inserted from lateral to medial with a twisting motion to open the sinus tarsi, to dilate the tarsal canal and to stretch the interosseous talocalcaneal ligament until the soft tissue inferior to the medial malleolus was tented. The cannulated probe was later replaced by the alignment rod, where the trial sizer and subtalar implant (BIOARCH or STA-Peg subtalar implant system of Wright medical group) were inserted. The size and position of the trial sizer and subtalar implant were verified via intraoperative C-arm fluoroscopy, where the leading edge of the implant should be 1/3 to 1/2 the distance across the subtalar joint. Before the end of the operation, the foot was manipulated to ensure the excessive pronation had been adequately corrected with a normal range of motion. The identical procedures were performed again on the contralateral foot. Different subtalar implant systems were used in our series because STA-Peg was no longer available and was replaced by BIOARCH in our institute after March 2016; both types of implants are effective for pediatric flexible flatfoot, but the latter was considered better in terms of radiographic correction [12]. The short leg walking cast was applied with weight-bearing as tolerated for two to three weeks, and orthotic insoles were used for the following 3 months to maintain the correction.

### 2.3. Outcomes Analyses

The functional assessment was done preoperatively and at the latest follow-up clinic visit (in a range of 33–83 months postoperatively) through the American Orthopaedic Foot and Ankle Society ankle–hindfoot scale (AOFAS-AHS), which quantifies pain, function and alignment. The radiographic assessment was done preoperatively and at each follow-up clinic visit (3 and 6 months postoperatively and once per year until skeletal maturity). The authors were blinded to the functional and perioperative data at the time of the radiograph review. The anterior–posterior and lateral weight-bearing radiographs of the foot were performed for the measurement of the following parameters (Figure 2): talonavicular coverage (the angle between the articular surfaces of the talus and that of the navicular on AP views; normal range <7°), talocalcaneal angle (the long axis of the talus intersects the lateral surface of the calcaneus on AP views; normal range 15–30°), talar–first metatarsal angle (the long axis of the talus intersects that of the first metatarsal on lateral weight-bearing views; normal range between 4° convex downward and 4° convex upward), talar declination angle (the long axis of the talus intersects the supporting surface on lateral views; normal is approximately 21°), lateral talocalcaneal angle (the long axis of the talus intersects the plantar border of the calcaneus on lateral views; normal range 25–45°), longitudinal arch angle (the plantar border of the calcaneus intersects the inferior edge of the fifth metatarsal on lateral views; normal range 150–170°), and calcaneal pitch (the plantar border of the calcaneus intersects the supporting surface on lateral views; normal range 20–30°).

### 2.4. Statistical Analyses

The results were expressed as medians and ranges. For statistical purposes, Wilcoxon signed-rank tests were assessed. Significance level was defined as *p* < 0.05.

## 3. Results

### 3.1. Surgical Outcome

Six patients (12 feet) underwent the index procedures and recovered uneventfully with the demographics shown in Table 1. The median operation time was 40 min per patient (with a range of 16–65 min). Removal of the implants was not required at latest follow-ups. There were no complications such as infection, sinus tarsi pain or implant extrusion. Only one patient suffered from humeral fracture 3 years postoperatively due to an accidental fall, which is believed to be not relevant to the SA. The functional and radiographic assessments are summarized in Table 2. The median AOFAS-AHS improved from 68 (with a range of 38–80) to 95 (range 71–97), and all of them were freely ambulatory without crutches. Both of the functional assessments and all parameters on the weight-bearing radiographs showed maintained improvement postoperatively with statistical significance (*p* < 0.05), and the adjacent bony surfaces to implants remained intact for all patients in our series.

### 3.2. Case Presentation

A representative case is as follows (case 5 in Table 1): an 11-year-old girl with Sillence type I OI presented with moderate pain in bilateral feet with calf muscle strain and plantar callosities since she was eight (Figure 3). Conservative treatments, including orthotic insoles and physical therapy, failed to improve the symptoms. A total of four fractures on the upper limbs were reported with minimal injuries. Physical examination showed bilateral flexible flatfoot, hindfoot valgus and gastrocnemius tightness with little or no soleus involvement. A range-of-motion test showed excessive pronation of the foot with extreme mobility at the subtalar and midfoot joints. The preoperative weight-bearing radiographs indicated severe flatfoot with subluxations at the subtalar and midfoot joints. At a 40-month postoperative follow-up, the symptoms completely resolved and the corrected alignment along with the position of the implants were maintained. The adjacent bony surfaces to implants remained intact, and there was no further fracture episode reported during the period of follow-up.

## 4. Discussion

Orthopedic surgeons are cautious in performing operative management on OI patients because of the nature of the disease, which is also true when dealing with symptomatic flexible flatfoot in OI patients. The bone fragility makes arthrodesis or osteotomies challenging since poor results have been shown in some case reports [5,6]. There is no definite guideline for the treatment of symptomatic flexible flatfoot in OI patients. Arthroereisis is a less invasive and non-fusion procedure that has shown minimal tissue trauma and faster recovery with comparable results [10,11], which may benefit OI patients in dealing with the flatfoot.

In our study, the median AOFAS-AHS improved from 68 (with a range of 38–80) to 95 (range 71–97). The improvement was mainly attributed to the corrected foot alignment and the relieved pain scale, where most patients reported occasional mild pain to painlessness from moderate daily pain preoperatively. There was no complication or re-operation performed. Only one patient suffered from a humeral fracture after the operation, and no other fracture episode was reported. The included patients were freely ambulatory without using crutches throughout the follow-up period. With appropriate operative management of the symptomatic flatfeet, the ambulatory function of OI type I patients was maintained if not improved and pain was reduced.

The radiographic parameters all improved significantly during the follow-up period in our series. The gaps between the radiographic change, morphological alternation and functional improvement of flatfoot patients have been widely studied. Indino et al. [8] showed that the correction of radiographic parameters was maintained not only at a short-term follow up, but also at mid-term follow up until skeletal maturity. However, they demonstrated that the amount of the morphologic correction at the end of the foot growth should be expected for lateral tarso-metatarsal alignment and talo-navicular congruency. The calcaneal parameters (calcaneal inclination and talocalcaneal angle) show a significant—in radiographic analysis—but small improvement that probably is not clinically relevant. In a pedobarographic study, Hagen et al. [13] further manifested that the morphologic correction after SA, such as decreased hindfoot pronation and forefoot supination, shifts the medially displaced load distribution to the lateral foot areas, normalizes the foot motion during gait and relieves foot discomfort. In a functional study through gait analysis and EMG study, Caravaggi et al. [14] showed that the activation of tibialis anterior muscle and gastrocnemius muscle were more physiological after SA. These findings at least partially support the effects of SA on improving the biomechanics of both feet and lower limbs.

A critical review of the literature on SA by Metcalfe et al. [10] reported that most complications in SA were attributed to malposition of the implant resulting in the under or overcorrection of the pronated foot, persistent discomfort from sinus tarsi pain and the migration or even extrusion of the implant. The complication rates range between 4.8% and 18.6% with unplanned removal rates between 7.1% and 19.3% across all implant types. Implant removal has been recommended with good results for sinus tarsi pain, and thus the rate of implant removal has been reported to be up to 39% in pediatric patients and even higher in adult patients [15]. Neil et al. [16] reported that most members of the AOFAS routinely remove the implants; however, the sinus tarsi pain was not a problem in our series due to the reduced mobility of OI patients, and removal of the implants was not required during the period of follow-up. On the other hand, considering the bone fragility of OI patients, the migration or even extrusion of the implant resulting in erosion of bone surfaces adjacent to the implants was once the major concern in our series. The reported complication rate due to implant extrusion ranges between 0.5% and 9% across all implant types [10], where pediatric patients with a higher BMI need to be handled cautiously due to their higher implant extrusion rate and worse outcomes [17]. However, the postoperative and follow-up weight-bearing radiographs showed that adjacent bony surfaces to implants remained intact, which could be attributed to less osteochondral damage of SA and the reduced mobility of OI patients. In addition to common complications after SA, Kumar et al. [18] reported a case that showed that talar neck fracture is a rare but devastating complication of SA, especially for those patients who have the change to participate in impact sports. Surgeons may have more concerns when performing SA in OI patients. In our series, no patient experienced talar or calcaneal fracture at latest follow-ups. Moreover, compared with the months of immobilization after arthrodesis or osteotomies, full weight-bearing as tolerated was possible immediately after the SA. Bone fragility may be aggravated by osteoporosis subsequent to immobilization [19], which falls into a vicious cycle of fracture, immobilization, osteoporosis and refracture. Considering the reduced osteochondral damage and faster recovery along with early full weight-bearing, SA seems to be appropriate for OI patients.

The role of the Vulpius procedure in conjunction with SA must be elaborated. A primary or secondary gastrocnemius equinus deformity is usually presented with flatfoot deformity [20] and addressed with an adjuvant soft tissue release procedure. Cicchinelli et al. [21] discussed the effectiveness of gastrocnemius recession with subtalar arthroereisis by comparing the radiographic outcomes of 28 feet in 20 pediatric patients with flatfoot in three surgical option groups: SA alone (8°, range −1~14°), SA with gastrocnemius recession (19°, range 11~34°), and SA with gastrocnemius recession and medial column stabilization (4°, range −9~19°). Although the study focused simply on the improvements in the radiographic measurements without further investigation of the possible correlated functional counterparts, these results demonstrated that gastrocnemius recession with SA has a statistically significant effect on the degree of radiographic correction of transverse plane elements of flatfoot.

To the best of our understanding, we provided the first case series of the treatment of symptomatic flatfoot in type I OI patients with SA. Some limitations should be considered, including the retrospective study design, the small sample size, relatively short follow-up, heterogeneous implants and lack of a control group. However, long-term follow-up and prospective multi-centered clinical trials at a larger scale may be relatively difficult due to the rarity of the disease, although these should be conducted to provide stronger evidence. Our series included exclusively skeletally immature patients, while the operative indication for similar conditions in adult patients remains to be explored. Despite these limitations, our study offers some preliminary information for pediatric OI patients with symptomatic flatfoot.

## 5. Conclusions

Owing to the rarity of the disease and bone fragility of OI, there remain severely limited experiences in surgical treatment for symptomatic flatfoot in OI patients. This is the very first study to investigate the therapeutic effects of SA and the Vulpius procedure on the safety and feasibility of treating this population. Considering the reduced osteochondral damage and faster recovery along with early full weight-bearing, SA seems to be appropriate for OI patients. Our data suggest that SA provides maintained radiographic correction and functional improvement, especially regarding pain relief without significant complications during the follow-up period. Further prospective, case-controlled studies are warranted.

## Figures and Tables

**Figure 1 ijerph-18-00067-f001:**
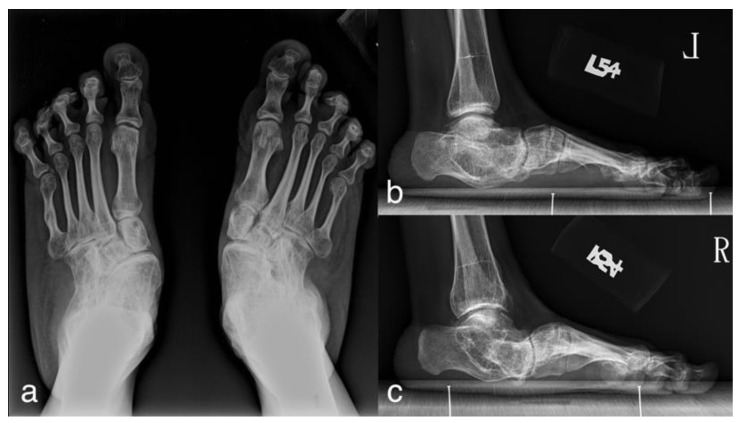
A 52-year-old woman with Sillence type I osteogenesis imperfecta (OI) and symptomatic flatfoot showing severe foot deformity. (**a**) AP radiograph; (**b**,**c**) lateral radiograph of left and right feet.

**Figure 2 ijerph-18-00067-f002:**
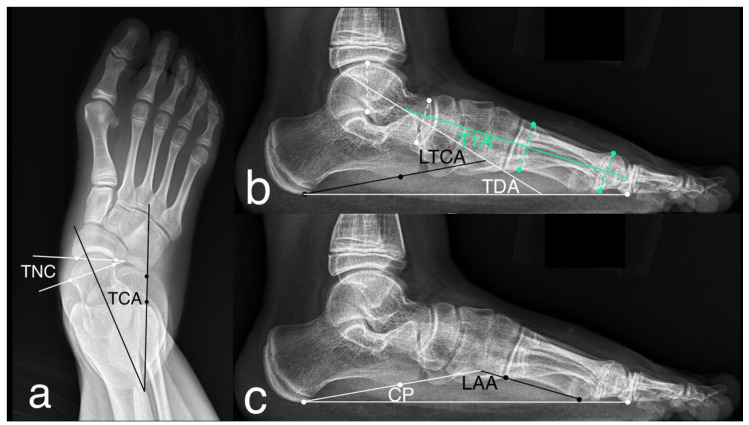
The radiographic measurements for flatfoot on anterior–posterior (AP) and lateral weight-bearing radiographs. (**a**) Talonavicular coverage (TNC) and talocalcaneal angle (TCA) on AP view; (**b**) talar–first metatarsal angle (T1A), talar declination angle (TDA) and lateral talocalcaneal angle (LTCA) on lateral view; (**c**) longitudinal arch angle (LAA) and calcaneal pitch (CP) on lateral view.

**Figure 3 ijerph-18-00067-f003:**
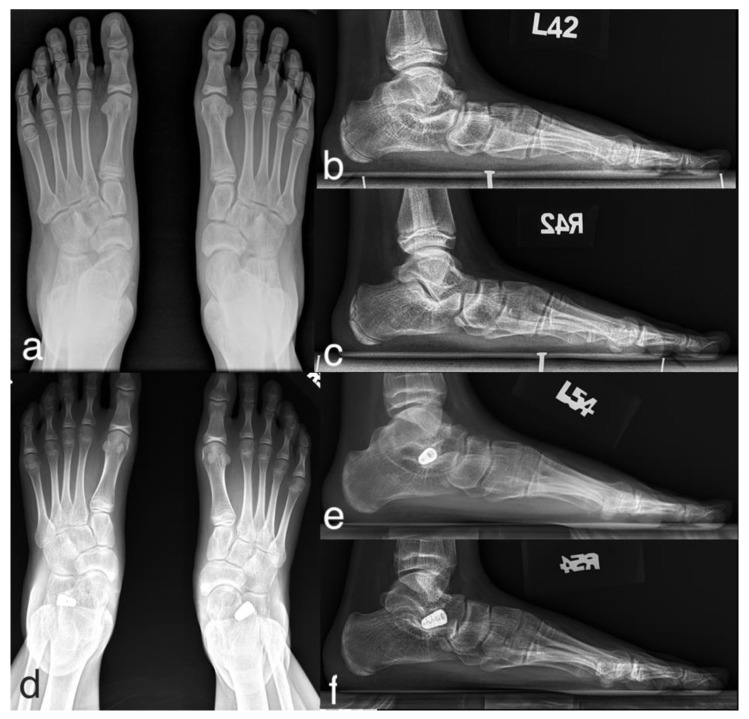
An 11-year-old girl with type I OI and symptomatic flexible flatfoot. The preoperative radiographs showed severe flatfoot deformity, which was significantly improved in 2-year postoperative follow-up radiographs. (**a**) Preoperative AP radiograph; (**b**,**c**) preoperative lateral radiograph of left and right feet; (**d**) postoperative AP radiograph; (**e**,**f**) postoperative lateral radiograph of left and right feet during 40-month follow-up.

**Table 1 ijerph-18-00067-t001:** The demographics for the included six patients.

Case	Gender	Age at the Time of Operation (Years)	Follow-up (Months)	Subtalar Implant	Preoperative AOFAS-AHS	Postoperative ^ AOFAS-AHS
1	F	11	83	STA-Peg	58	71
2	F	11	61	STA-Peg	38	79
3	M	9	58	STA-Peg	66	97
4	F	5	54	BIOARCH	77	94
5	F	11	40	BIOARCH	80	97
6	F	7	33	BIOARCH	70	97

^ at the latest follow up clinic (33–83 months postoperatively). AOFAS-AHS = American Orthopaedic Foot and Ankle Society Ankle–Hindfoot Scale.

**Table 2 ijerph-18-00067-t002:** The functional and radiographic assessments for patients undergoing SA surgery.

Evaluation Parameters			Preoperative	Postoperative ^
Functional Assessment		AOFAS-AHS	68 (38–80)	95 (71–97) *
Radiographic Assessments	AP	Talonavicular coverage	38 (25–48)	15 (5–25) *
		Talocalcaneal angle	35 (18–50)	27 (19–32) *
	Lateral	Talar-first metatarsal angle	29 (16–39)	10 (1–22) *
		Talar declination angle	42 (26–53)	28 (21–34) *
		Lateral Talocalcaneal angle	52 (33–62)	39 (28–48) *
		Longitudinal arch angle	168 (166–173)	166 (161–172) *
		Calcaneal pitch	10 (8–15)	12 (7–16) *

* *p*-value < 0.05, Wilcoxon signed-rank test. ^ at the latest follow up clinic (33–83 months postoperatively). AOFAS-AHS = American Orthopaedic Foot and Ankle Society Ankle–Hindfoot Scale. AP = anterior–posterior. SA = subtalar arthroereisis.

## Data Availability

The data presented in this study are available on request from the corresponding author. The data are not publicly available due to restrictions of patient privacy and ethics.

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
