# Peer review of "Preliminary Results of Subtalar Arthroereisis with Vulpius Procedure for Symptomatic Flatfoot in Patients with Type I Osteogenesis Imperfecta"

_ijerph, 2020, doi:10.3390/ijerph18010067_

Round 1
Reviewer 1 Report
Dear Editor,
Thanks for the oportunity to review the manuscript entitled " Preliminary Results of Subtalar Arthroereisis with Vulpius procedure for Symptomatic Flatfoot in Patients with Type I Osteogenesis Imperfecta" which aims to assess the safety and feasibility of adapting subtalar arthroereisis (SA) for type I osteogenesis imperfecta (OI) patients with symptomatic flatfoot.
I recommend to the authors to addressed some issues:
Introduction:
- Rationally is not clear.
- Literature should be updated with recent articles.
Methods:
- Type of study in not clear
- How the authors have selected the sample and what was the criteria for this sample size?
Results:
Discussion:
References:
- Articles from IJERPH should be included int he text.
- The manuscript lacks an update. Just one reference is from the last 5 years.
Author Response
Response to Reviewer 1 Comments
Point 1: Rationally is not clear.
Response 1: Thank you for the helpful suggestion. The introduction section was revised rationally in Line 67-85, and was summarized as followed.
“Symptomatic flatfoot deformity in Sillence type I OI patients is often overlooked for therapeutic intervention because of their relatively high functional status and the possible unsatisfactory outcomes by other interventions reported previously. The aim of our study was to improve the clinical symptoms and to postpone the deterioration of ambulatory function of these patients by a relatively less traumatic intervention. Therefore, we conducted a retrospective study for symptomatic flatfoot in Sillence Type I OI patients, to assess the safety, feasibility and the clinical results of SA for this challenging problem.”
Point 2: Literature should be updated with recent articles.
Response 2: Thank you for the helpful suggestion. The reference list was supplemented with eight up-to-date research as stated below.
Point 3: Type of study is not clear
Response 3: Thank you for the attentive suggestion. The type of study is a retrospective case-series.
Point 4: How the authors have selected the sample and what was the criteria for this sample size?
Response 4: Thank you for the important comment, where we failed to explain clearly. The inclusion criteria were the symptomatic flatfoot of skeletally immature and ambulatory Sillence type I OI patients who had failed conservative treatments. It was revised in Line 88-94 as followed.
“With the approval of the Research Ethics Committee from our institute (201909005RINB), we conducted a retrospective study to review the 35 OI patients with symptomatic flatfoot deformity in our institute. Skeletally matured patients, non-ambulators or non-Sillence-type-I patients were excluded. Six patients (5 girls and 1 boy, 12 feet) with symptomatic flexible flatfoot treated with bilateral SA and Vulpius procedure between December 2013 and January 2018 in our institute were identified. All of them had failed conservative managements including orthotic insoles, physical therapy and activity modification for at least 6 months before the consideration for surgical treatment.”
Point 5: Articles from IJERPH should be included in the text.
The manuscript lacks an update. Just one reference is from the last 5 years.
Response 5: Thank you for the valuable and important suggestion. We would really like to include articles from IJERPH, but limited research about OI or flatfoot can be listed. Due to the rarity of OI, the reference study was scare. However, the recent discovery on subtalar arthroereisis with an overall review of OI were added and discussed in Line 37-39/ 74-81 (introduction section), Line 133-137 (material and methods section) and Line 231-246/ 258-260(discussion section) as followed.
- Marini, J.C. et al. Osteogenesis imperfecta. Rev. Dis. Prim. 2017, 3, 17052.
https://pubmed.ncbi.nlm.nih.gov/28820180/ DOI: 10.1038/nrdp.2017.52 - Bernasconi, A. et al. Midterm assessment of subtalar arthroereisis for correction of flexible flatfeet in children. Traumatol. Surg. Res. 2020, 106, 185–191.
https://pubmed.ncbi.nlm.nih.gov/31848065/ DOI: 10.1016/j.otsr.2019.10.012
- Indino, C. et al. Effectiveness of subtalar arthroereisis with endorthesis for pediatric flexible flat foot: a retrospective cross-sectional study with final follow up at skeletal maturity. Foot Ankle Surg. 2020, 26, 98–104.
https://pubmed.ncbi.nlm.nih.gov/30598422/ DOI: 1016/j.fas.2018.12.002 - Tan, J.H.I. et al. The outcomes of subtalar arthroereisis in pes planus: a systemic review and meta-analysis. Orthop. Trauma Surg. 2020.
https://pubmed.ncbi.nlm.nih.gov/32377845/ DOI: 10.1007/s00402-020-03458-8 - Hagen, L.et al. Pedobarographic changes during first month after subtalar extra-articular screw arthroereisis (SESA) operation of juvenile flexible flatfoot. Orthop. Trauma Surg. 2020, 140, 313–320.
https://pubmed.ncbi.nlm.nih.gov/31321498/ DOI: 10.1007/s00402-019-03230-7 - Caravaggi, P. et al. A. Functional evaluation of bilateral subtalar arthroereisis for the correction of flexible flatfoot in children: 1-year follow-up. Gait Posture 2018, 64, 152–158.
https://pubmed.ncbi.nlm.nih.gov/29909229/ DOI: 1016/j.gaitpost.2018.06.023 - Hsieh, C.H. et al. Endosinotarsal device exerts a better postoperative correction in Meary’s angle than exosinotarsal screw from a meta-analysis in pediatric flatfoot. Rep. 2020, 10, 1–11.
https://pubmed.ncbi.nlm.nih.gov/32782334/ DOI: 10.1038/s41598-020-70545-6 - Hsieh, C.H. et al. Body Weight Effects on Extra-Osseous Subtalar Arthroereisis. 2019, 1–10.
https://pubmed.ncbi.nlm.nih.gov/31443407/ DOI: 3390/jcm8091273
Reviewer 2 Report
The topic is original, interesting, and the manuscript is well written. Despite this, some minor modifications are needed.
Line 64-66: subtalar arthoreisis showed an improvement of some specific radiographic parameters in pediatric populations that better explain its potential power of correction better than a general “reduction of a flatfoot deformity”: I suggest to cite these parameters (DOI: 10.1016/j.fas.2018.12.002).
Operative technique. Were the procedures performed bilaterally at the same surgery? If not, how long was the time interval between the surgeries?
Line 103: the removal of implants should be moved in the results section
Line 111-114: a short description, a normal range and an image of the radiographic parameters should be provided.
Line 114-115: statistical methods should be part of a separate subsection (statistical analysis).
Line 120: how was the blood loss measured? If no data are available I suggest to remove the blood-loss sentence even because it is not necessary since the procedure is supposed to be not at risk of high blood loss.
Line 121: is the humeral fracture correlated to the procedure described? How it is correlated? Please, provide an explanation.
Table 2: is the p-value of the last follow-up parameters related to the per-operative parameters or to the 2-years-follow-up parameters? In methods section it is not declared a mid-term follow-up endpoint so why there is the 2-years follow-up parameters? I suggest to remove the 2-year column that is confounding.
Author Response
Response to Reviewer 2 Comments
Point 1: Line 64-66: subtalar arthoreisis showed an improvement of some specific radiographic parameters in pediatric populations that better explain its potential power of correction better than a general “reduction of a flatfoot deformity”: I suggest to cite these parameters (DOI: 10.1016/j.fas.2018.12.002).
Response 1: Thank you very much for the valuable comment. The suggested article is so informative guided us to other great references. It was added and discussed thoroughly in Line 78-79 (introduction section) and Line 233-240 (discussion section).
Point 2: Operative technique. Were the procedures performed bilaterally at the same surgery? If not, how long was the time interval between the surgeries?
Response 2: Thank you for the attentive comment. The procedures were performed bilaterally at the same surgery. The information was added in Line 115 and 133.
Point 3: Line 103: the removal of implants should be moved in the results section
Response 3: Thank you for the valuable suggestion. It was shifted to Line 172 in the results section.
Point 4: Line 111-114: a short description, a normal range and an image of the radiographic parameters should be provided.
Response 4: Thank you for the considerate suggestion. Considering the possible readers of IJERPH, providing information to radiographic parameters is necessitated. New figure 2 was added and a short description with normal range was provided on line 146-163.
Point 5: Line 114-115: statistical methods should be part of a separate subsection (statistical analysis).
Response 5: Thank you for the valuable suggestion. It was shift to Line 164-166.
Point 6: Line 120: how was the blood loss measured? If no data are available, I suggest to remove the blood-loss sentence even because it is not necessary since the procedure is supposed to be not at risk of high blood loss.
Response 6: Thank you for the attentive comment. The blood-loss was measured through estimation on the weight of gauze. However, we totally agreed that it was unnecessary to state, and it was removed.
Point 7: Line 121: is the humeral fracture correlated to the procedure described? How it is correlated? Please, provide an explanation.
Response 7: Thank you for the valuable suggestion. The fracture episode occurred 3-year postoperatively due to an accidental fall, and it is believed to be not relevant to the SA operation. The information was added on line 174.
Point 8: Table 2: is the p-value of the last follow-up parameters related to the per-operative parameters or to the 2-years-follow-up parameters? In methods section it is not declared a mid-term follow-up endpoint so why there is the 2-years follow-up parameters? I suggest to remove the 2-year column that is confounding.
Response 8: Thank you for the helpful comment. Both p-value of the last follow-up parameters and the 2-years-follow-up parameters are related to the per-operative parameters. However, it is definitely confusing, so the 2-year column was removed as suggested in Line 186.
Reviewer 3 Report
Dear authors
The study you presented is interesting and well-written, with scientific soundness. The authors claim that this manuscript is one of the first reports for pediatric OI patients with symptomatic flatfoot. Despite its limitation, the authors are aware of them, and future work in this approach is promising. After I read the manuscript, I clearly understand the case report and its particularity. In my point of view, the study is properly discussed and no further comments are given on this occassion.
My unique complaint, its that references are scare and old. Understanding that this type of disorder it's rare, I understand, that this reference list cannot be improved
Please revise if these references can be used to improve your reference list and discussion your manuscript
https://pubmed.ncbi.nlm.nih.gov/29909229/
https://www.ncbi.nlm.nih.gov/pmc/articles/PMC4621198/
https://pubmed.ncbi.nlm.nih.gov/31474401/
https://pubmed.ncbi.nlm.nih.gov/32582390/
https://journals.sagepub.com/doi/abs/10.1177/1071100715604237
https://pubmed.ncbi.nlm.nih.gov/27492436/
Author Response
Response to Reviewer 3 Comment
Point: Please revise if these references can be used to improve your reference list and discussion your manuscript
https://pubmed.ncbi.nlm.nih.gov/29909229/
https://www.ncbi.nlm.nih.gov/pmc/articles/PMC4621198/
https://pubmed.ncbi.nlm.nih.gov/31474401/
https://pubmed.ncbi.nlm.nih.gov/32582390/
https://journals.sagepub.com/doi/abs/10.1177/1071100715604237
https://pubmed.ncbi.nlm.nih.gov/27492436/
Response: Thank you for the considerate comment and suggestion. The first reference provided was so informative and suitable for our manuscript. It was added and discussed thoroughly on Line 233-240 in the discussion section.
- Indino, C. et al. Effectiveness of subtalar arthroereisis with endorthesis for pediatric flexible flat foot: a retrospective cross-sectional study with final follow up at skeletal maturity. Foot Ankle Surg. 2020, 26, 98–104.
https://pubmed.ncbi.nlm.nih.gov/30598422/ DOI: 1016/j.fas.2018.12.002
As for the other provided references, they guided us to the latest reported studies and added to the reference list as below. Thank you very much!
- Marini, J.C. et al. Osteogenesis imperfecta. Rev. Dis. Prim. 2017, 3, 17052.
https://pubmed.ncbi.nlm.nih.gov/28820180/ DOI: 10.1038/nrdp.2017.52 - Bernasconi, A. et al. Midterm assessment of subtalar arthroereisis for correction of flexible flatfeet in children. Traumatol. Surg. Res. 2020, 106, 185–191.
https://pubmed.ncbi.nlm.nih.gov/31848065/ DOI: 10.1016/j.otsr.2019.10.012 - Tan, J.H.I. et al. The outcomes of subtalar arthroereisis in pes planus: a systemic review and meta-analysis. Orthop. Trauma Surg. 2020.
https://pubmed.ncbi.nlm.nih.gov/32377845/ DOI: 10.1007/s00402-020-03458-8 - Hagen, L.et al. Pedobarographic changes during first month after subtalar extra-articular screw arthroereisis (SESA) operation of juvenile flexible flatfoot. Orthop. Trauma Surg. 2020, 140, 313–320.
https://pubmed.ncbi.nlm.nih.gov/31321498/ DOI: 10.1007/s00402-019-03230-7 - Caravaggi, P. et al. A. Functional evaluation of bilateral subtalar arthroereisis for the correction of flexible flatfoot in children: 1-year follow-up. Gait Posture 2018, 64, 152–158.
https://pubmed.ncbi.nlm.nih.gov/29909229/ DOI: 1016/j.gaitpost.2018.06.023
Besides, two of the studies from our team were added for better understanding of SA in Line 136 & 260.
- Hsieh, C.H. et al. Endosinotarsal device exerts a better postoperative correction in Meary’s angle than exosinotarsal screw from a meta-analysis in pediatric flatfoot. Rep. 2020, 10, 1–11.
https://pubmed.ncbi.nlm.nih.gov/32782334/ DOI: 10.1038/s41598-020-70545-6 - Hsieh, C.H. et al. Body Weight Effects on Extra-Osseous Subtalar Arthroereisis. 2019, 1–10.
https://pubmed.ncbi.nlm.nih.gov/31443407/ DOI: 3390/jcm8091273
Round 2
Reviewer 1 Report
The authors have addressed all my comments.
However, I think that this manuscript in some parts lacks of scientific support from the literature and also lacks of in depth discussion. Only 20 references have been used in all manuscript, and several are out of date. These aspect should be taken in account by the Editor.